# The impact of psychological theory on the treatment of Attention Deficit Hyperactivity Disorder (ADHD) in adults: A scoping review

Rebecca E. Champ[1]◐*, Marios Adamou[1]‡, Barry Tolchard[2]‡

**1** Department of Nursing and Midwifery, School of Human and Health Sciences, University of Huddersfield, Huddersfield, United Kingdom, **2** School of Health and Life Sciences, Teeside University, Middlesbrough, United Kingdom

◐ These authors contributed equally to this work.
‡ These authors also contributed equally to this work.
* Rebecca.Champ@hud.ac.uk

**Data Availability Statement:** All relevant data are within the paper and its Supporting Information files.

## Abstract

Psychological theory and interpretation of research are key elements influencing clinical treatment development and design in Attention Deficit Hyperactivity Disorder (ADHD). Research-based treatment recommendations primarily support Cognitive Behavioural Therapy (CBT), an extension of the cognitive behavioural theory, which promotes a deficit-focused characterisation of ADHD and prioritises symptom reduction and cognitive control of self-regulation as treatment outcomes. A wide variety of approaches have developed to improve ADHD outcomes in adults, and this review aimed to map the theoretical foundations of treatment design to understand their impact. A scoping review and analysis were performed on 221 documents to compare the theoretical influences in research, treatment approach, and theoretical citations. Results showed that despite variation in the application, current treatments characterise ADHD from a single paradigm of cognitive behavioural theory. A single theoretical perspective is limiting research for effective treatments for ADHD to address ongoing issues such as accommodating context variability and heterogeneity. Research into alternative theoretical characterisations of ADHD is recommended to provide treatment design opportunities to better understand and address symptoms.

## Introduction

The combination of psychological theory and interpretation of research have been highlighted as critical influencers guiding decision-making for clinical treatment design and development for Attention Deficit Hyperactivity Disorder (ADHD) [1, 2]. ADHD is a neurodevelopmental disorder of self-regulation with symptoms negatively affecting daily functioning at work and at home, with long-term impacts in academic, occupational, social and emotional areas of functioning [3–8]. Effective, long-term treatment outcomes benefit both the individual with ADHD and society as a whole as undiagnosed and untreated adults with ADHD may become an economic burden due to increased health care costs and decreased productivity at work [9, 10].

**Funding:** The author(s) received no specific funding for this work.

**Competing interests:** The authors have declared that no competing interests exist.

Russell Barkley [11] postulated the first unifying theory of ADHD, which places a core deficit of behavioural inhibition at the source of ADHD behaviours. Several theoretical models attribute additional and alternative cognitive sources for the development of ADHD symptoms [12–14]. While a variety of different interventions are available and the benefit of other forms of support is acknowledged (e.g. psychotherapy or coaching), only Cognitive Behavioural Therapy (CBT), Mindfulness, Dialectical Behavioural Therapy (DBT) and potentially Neurofeedback have the most empirical support [15]. Results of non-pharmacological intervention studies suggest these interventions have a positive effect on core behavioural symptoms of ADHD (inattention, hyperactivity/impulsivity), particularly when compared to inactive control conditions [15, 16]. However, recent systematic reviews of non-pharmacological treatment highlight that different classes of intervention design take similar approaches; that heterogeneity in sample size, study design, quality and symptom outcome measurement makes meta-analysis difficult, and there is a high risk of bias [15–17]. Additionally, the National Institute for Health and Care Excellence (NICE) [18] only recommends interventions that match a similar protocol to medications: Randomised Controlled Trials (RCTs), primarily based in CBT [15], despite a growing wider evidence base.

It is hypothesised that much of current research for the characterisation of ADHD is based on a cognitive behavioural theoretical paradigm that does not account comprehensively for the broad spectrum of ADHD presentation [1, 19–24]. This paradigm is deficit-focused with primary treatment outcomes of symptom reduction and control of maladaptive behaviours. Recent research in psychology suggests that this may not be the best approach to improving mental health, and it may be necessary to develop positive psychological factors and emotions that cultivate health and wellbeing [25, 26]. This scoping review aims to map the evidence and understand the influence of current psychological theories on design and treatment recommendations in adult ADHD by answering the following questions:

1. Are characterisations of ADHD dominated by a cognitive behavioural paradigm?

2. Does that paradigm influence treatment design and outcomes?

3. Are there any alternative characterisations of ADHD that present a different perspective to the cognitive behavioural paradigm?

A broad approach was considered most effective to identify gaps in the literature, as data regarding supportive psychological theories would likely be identified in publications beyond specific study designs. To our knowledge, this is the first scoping review providing an overview of the theoretical characterisations of ADHD and their impact on available treatments.

## Methods

### Search strategy

The scoping review was carried out over three months: February, March and April 2020. The scoping review protocol was published on the Open Science Framework (https://osf.io/). Search design and criteria were formulated based on guidance and recommendations by Arksey & O'Malley [27], Colquhoun et al. [28], O'Brien et al. [29] and the Joanna Briggs Institute [30]. A starting timeframe from the publication of Barkley's [11] theory was selected as the foundation for current theoretical characterisations of ADHD. Papers were reviewed from multiple countries, including the United States, the United Kingdom, The Netherlands, Canada, Argentina, Brazil, Colombia, Iceland, Ireland, Portugal, Spain, Belgium, Germany, Switzerland, Finland, Sweden, Israel, Iran, China, Hong Kong, India and Australia, and multiple languages including English, Dutch, German, French, and Spanish.

**Table 1. Search strategy.**

| Search Combinations | Resources |
|---|---|
| **Primary search string:** | **Specific Databases:** |
| Adult OR Student OR College OR University | CINHAL |
| AND | PsycInfo |
| ADHD OR Attention Deficit Hyperactivity Disorder | PubMed |
| AND | SCOPUS |
| Treat* OR Counsel* OR Manag* | |
| Filtered for Age: 19+ | |
| **Combinations:** | |
| Treat* OR Counsel* OR Manag* NOT (child* OR Parent OR Drug OR stimulant OR pharma*) | |
| Therap* NOT child* OR Parent NOT Drug OR stimulant OR pharma* | |
| Coaching | |

Research evidence was identified by conducting searches across web-based databases with pre-determined search terms. Table 1 outlines the search terms and syntax used in primary and secondary searches.

Additional searches were in generic search engines Google and Google Scholar, and checks of references from guidance documents and systematic reviews for additional material. Once identified, these references were collected through additional database searches or a direct search in the specific journal or publication.

## Inclusion criteria

Titles and abstracts of materials were reviewed for eligibility. Materials were considered appropriate if they met the following criteria:

- Studies involving research on a pilot, efficacy, or applicability of a treatment intervention for adults with ADHD (19–65+, male and female)

- Systematic reviews of treatment literature or specific approaches to treatment for adults with ADHD

- Thesis, conference papers, or reports reviewing, presenting, or recommending treatment approaches for adults with ADHD

- Documents, articles, books, or consensus statements presenting guidance or recommendations for treatment for adults with ADHD

## Exclusion criteria

In addition to meeting the inclusion criteria, materials were excluded if they met one of the following exclusion criteria:

- Treatment approaches designed for a specific subset of participants (couples, military, substance abuse)

- Treatment approaches designed to treat specific comorbidities (Autism, Bi-polar, Learning Disabilities, Tourette's, Oppositional Defiant Disorder, Personality Disorder, Traumatic Brain Injury)

- Treatment approaches designed for the inclusion of younger age groups (children, adolescents) or their parents

- Materials summarising and updating recent developments in the field of treatment for adult ADHD (general practice journals, nursing practice journals, medical student journals)

- Characterisations of adult ADHD that were not empirically researched

A large body of literature has been published over the years which present different characterisations of adult ADHD and subsequent recommendations for treatment. Predominantly based in the US, these biopsychosocial models range from origin theories of genetic strengths [31], diversity [32] and developmental impairment of the prefrontal cortex due to issues with attachment and trauma [33], to identifying multiple presentations of ADHD diagnosed individually with SPECT imaging [34]. While these models do present alternative characterisations of ADHD, they are not empirically researched and therefore will be excluded from this review.

The following PRISMA flowchart (Fig 1) presents the search process details, including the number of articles located, those eliminated and those included in the final analysis.

## Results and analysis

The 221 articles were subdivided into the following categories according to their primary content: Cognitive Behavioural Therapy (122), Coaching (36), Psychotherapy (16), and Other

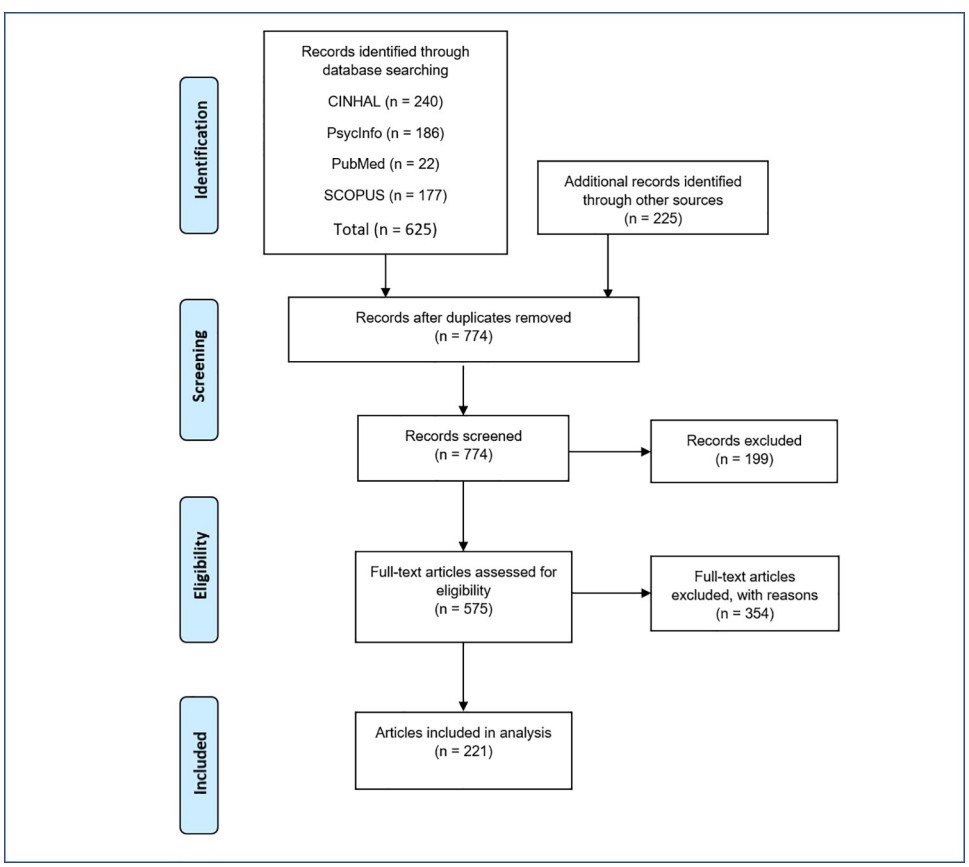

**Fig 1. Flowchart of scoping review: Characterisation of adult ADHD.**

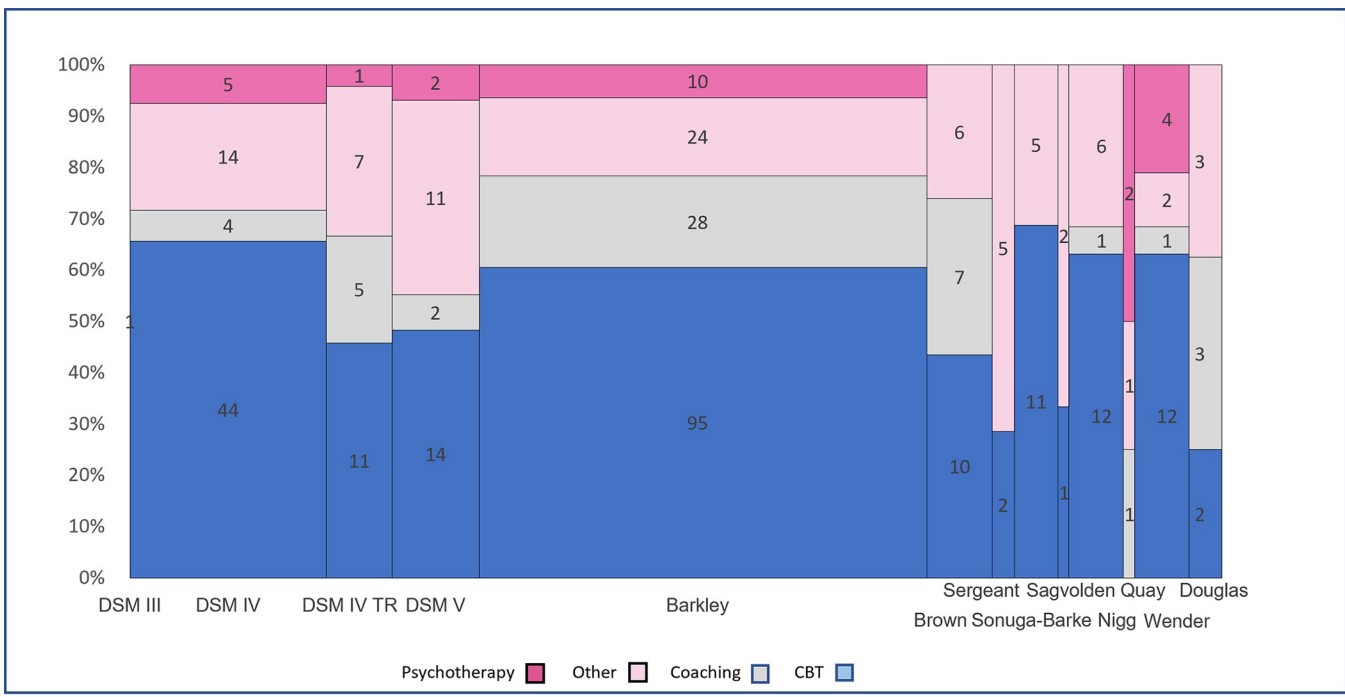

**Fig 2. ADHD characterisations cited in intervention category.**

(47). All articles were assessed for quality against the relevant Critical Appraisals Skills Programme (CASP) checklists. Results summary of the ADHD characterisation cited for each intervention category is displayed in a mosaic plot (Fig 2).

An overview of interventions published by year is displayed in a column chart (Fig 3).

Due to the number and variety of materials, a narrative analysis was performed to review the publications' composition. Systematic reviews were also analysed separately to see if any review of the characterisation of ADHD had been completed previously.

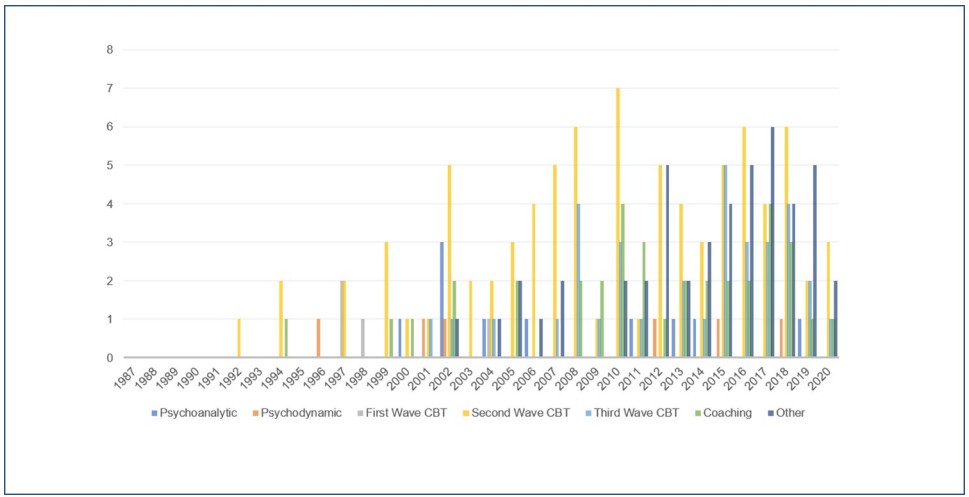

**Fig 3. ADHD interventions published by year.**

## Systematic reviews

Over the years, several systematic reviews of treatments for adult ADHD have been published. Searches were undertaken through Joanna Briggs Institute Evidence Synthesis, The Cochrane Database of Systematic Reviews, and the Campbell Library show some of these reviews are specific to the efficacy of a particular intervention approach, such as psychodynamic therapy [35], homoeopathy [36], Cognitive Behavioural Therapy (CBT) [37–41], Mindfulness [42, 43], and Meditation [44, 45]. Others have been focused on efficacy [2, 15, 16, 46–48], long term outcomes [10, 49], or guidance [50]. Only one systematic review investigated the characterisation of adult ADHD but failed to find evidence of neurocognitive disfunction as a predictor of persistence [51]. Interestingly, one systematic review investigated how adults with ADHD experience and manage their symptoms [52]. Therefore, it seems that no recent attempt has reviewed the psychological theories for the characterisation of adult ADHD.

## Data extraction

Articles were reviewed, and data extracted into categorised excel spreadsheets for comparison. Each document was examined for the following criteria:

1. Date

2. Authors

3. Research Purpose/Outcome

4. Treatment Approach

5. ADHD Characterisation

6. ADHD Theory

## Intervention analysis

Research studies and guidance documents present treatment approaches in various environments, contexts and skill levels of delivery. The following interventions present a wide range of delivery in clinical and non-clinical environments; therapeutic, academic, and social contexts; and professionals including psychiatrists, clinical psychologists, nurses, occupational therapists, psychotherapists, as well as counsellors, coaches, and mentors.

**Cognitive Behavioural Therapy (CBT).**   Due to the involvement of cognitive behavioural theory in establishing the characterisation of ADHD, the bulk of research in the field has used this intervention model. While recommended as the primary treatment modality, treatment goals and methodologies have changed over time. Due to this shift in focus, some early research references non-empirical anecdotal work. This analysis will use the delineation of "waves" as seen in the current theoretical literature to demonstrate these changes [53] (S1 Appendix).

*First wave*: *Behaviourism*. Before 1990, ADHD was still considered a disorder of childhood. Although considerable research exists regarding "first wave" treatments in children, the few approaches developed for adults apply pure behavioural theory and techniques. A case study of three subjects focused on improving attentional behaviour in psychiatric patients [54], and the design utilises operant theory and models used for brain injury [55]. Guidance documents for teachers, parents and counsellors [56] and psychotherapists [57] highlight the importance of behavioural skill development. The conceptualisation of ADHD in these treatment models is a disorder of attention [54] and a neurobiological disorder of self-regulation, executive

function deficits and disinhibition [56, 57]. Burgess et al. [54] exemplify the transition of the conceptualisation of attentional deficits in ADHD from mental illness to mental disorder.

The therapist's role in these models is to assist the client in learning and practising practical behavioural strategies for task attention, organisation, listening and scheduling, and planning and organising daily activities. Treatment designs are varied, with only one specifying weekly sessions from 6–10 weeks [54]. Characterisation of ADHD aetiology highlight issues with inability to maintain vigilance (attention deficit) and distractibility [54], self-regulation, impaired inhibition, developmental delay, and deficits in executive function, referencing Barkley [56, 57].

*Rational Emotive Behaviour Therapy (REBT)*. A single case study for ADHD specifically used REBT [58]. REBT approaches ADHD as a combination of neurobiological deficits [59] and developed secondary psychological personality disorders. Failure to develop cognitive structures leads to a lack of connection between thoughts, emotions and feelings, leading to deeply held distorted convictions and beliefs. The therapist's role in this model is to assist the individual to dispute evaluative cognitions ("musts") to develop a more rational philosophical orientation to the world. The approach incorporates independent "experiments" by clients outside of therapy, problem-solving methodology, and gentle introduction of rational self-statements for clients who lack the cognitive sophistication to engage in disputing of irrational beliefs [58, p. 95]. Treatment design has a developmental and longitudinal focus, in this case eight years. Characterisation of ADHD is described as DSM-IV core symptoms of attentional difficulties, impulsivity and hyperactivity [60] and references both Douglas's [61] cognitive processing deficit model and Barkley's [59] model of response inhibition and executive function deficits contributing to deficient self-regulation, impaired cross-temporal organisation of behaviour, and diminished social effectiveness and adaptation.

*Second wave*: *Cognitive and cognitive behavioural therapy model.* Cognitive and Cognitive Behavioural approaches are the primary and recommended treatment for working with ADHD and therefore make up the bulk of studies reviewed for this analysis. Considered "second wave" cognitive behavioural therapies, they consist of systematic reviews [2, 37, 38, 41], randomised controlled trials (RCT) [62–80], group interventions [81–87], individual interventions [88–90], quantitative analysis [91], qualitative analysis [92], a cohort study [93], case-control studies [69, 94–98], single case studies [99–102], multiple case studies [103–106], and psychotherapeutic treatment guidance [8, 106–138]. Many of these studies deliver CBT as a standalone intervention, however multimodal treatment is recommended, and several treatment models include individual coaching or mentoring support alongside or in between CBT sessions [57, 72, 77, 86, 103, 112, 127, 131, 137, 139]. One intervention also includes hypnosis and CBT [140].

Second wave interventions for ADHD recognise the neurobiological deficits as specific to the disorder and not brain injury. While they stress there is no "cure" for ADHD and the literature is clear that ADHD does not arise from distorted cognitions, cognitive treatment models focus primarily on improving, strengthening, or retraining cognitive abilities to increase awareness of behaviour and behavioural control. Early research identified cognitive distortions and maladaptive strategies and beliefs as interfering with skills acquisition and therefore needed support [104, 136]. Further research shifted this view slightly to perceive the development of a negative self-concept as the core issue for maladaptive schema or "secondary symptoms" of stress, anxiety, depression, and chronic perceived failure attributed to a history of unachieved potential and negative feedback resulting from a lack of recognition of the disorder. [122, 141, 142]. Aims of treatment reduce deficit-based symptoms, develop environmental restructuring and accommodations, improve self-esteem and negative self-concept through disorder psychoeducation, and increase confidence in capabilities through supported skills

practice and repetition. Approaches vary widely, including cognitive rehabilitation, cognitive and metacognitive remediation, and cognitive restructuring. However, most treatment approaches in this area are defined as Cognitive Behavioural Therapy (CBT) (S1 Appendix).

The therapists' role in these models is more a "partner", "expert teacher-motivator" [122] and collaborator than the traditional medical expert role [117, 121, 129]. Originally defined by Hallowell & Ratey [143] as "coaching", therapists are encouraged to be active and directive in providing structure and redirection to goals or session topics [83, 87, 101, 108, 129, 135, 137, 144]. Failure to initiate behavioural changes or maintain new habits and strategies, or "procrastivity", is attributed to motivational problems due to the nature of the disorder [8, 108, 121, 145]. CBT for ADHD identifies the ADHD client's difficulty with delayed gratification and generation of positive emotions as the reason for lack of engagement or "Coping Drift", where individuals stop implementing the skills taught in treatment [121, 145]. Professionals are cautioned that repetition is key, and strategies must be reinforced, or relapse is likely. Therefore relapse prevention is included in practice as well as model design [8, 70, 71, 74, 83, 86, 93, 94, 108, 121, 126, 136, 145, 146]. Recommendations for resistance or avoidance of aversive emotional states is to provide therapist support to develop tolerance [107, 122, 135], reframe past experiences [112, 128, 147], and build resilience when encountering setbacks [8, 111, 118, 137, 138, 146]. Treatment designs are limited in length, either by the number of sessions (3 to 16) or by relevance (academic year) except for single case studies [99–102, 104]. Intervention delivery methods vary from individual or group therapy and didactic teaching with therapeutic support to a computerised program and self-help manuals (S1 Appendix).

Characterisation of ADHD and aetiology highlight issues with attentional and behavioural control (hyperactivity, impulsivity, disorganisation) initially, but broadens to give a higher priority to executive dysfunction deficits, motivation and sustained attention, issues with emotional control and self-regulation. Guidance documents definitions of ADHD are often cited: of the 84 papers in this Second Wave analysis, 29 reference DSM-IV [60], seven reference DSM-IV-TR [148], and eight reference DSM-V [149]. Several studies reference alternative characterisations of ADHD, such as similarity to brain injury [117] and Brown's Executive Function model [83, 94]. However, Barkley is cited in 70 documents.

*Third Wave*: *Mindfulness and acceptance*. Third Wave cognitive behavioural interventions take a different treatment approach to traditional CBT. While they are similar in the practical application of behavioural techniques, they differ in their theoretical approach and the focus on cognitive change. Third-wave approaches explore context: the relationship between a person's thoughts and emotions rather than content alone. This relationship includes a more holistic perspective of health beyond the reduction of disorders [53]. Therefore, this analysis will review them separately. These approaches include Metacognitive Therapy (MCT), Dialectical Behavioural Therapy (DBT), and Mindfulness Cognitive Behavioural Therapy (MCBT).

*Metacognitive Therapy (MC)*. Four documents used a metacognitive approach (MC), divided into group metacognitive therapy [5, 76, 150] and metacognitive remedial psychotherapeutic guidance [151]. Metacognitive interventions conceptualise ADHD as neurobiological dysfunction in the corticostriatal pathways, displayed as deficits in executive functions [151]. MC highlights the importance of awareness of cognitions or thinking about thinking to strengthen executive functions to enhance functioning and improve self-control. Borrowing from the psychoanalytic frame [152], treatment of this hybrid model aims to develop an "observing ego" or self-awareness, increasing the ability to be conscious of maladaptive thoughts and behaviours and confront them via self-analysis. The therapist's role is to focus on cognitive and behavioural aspects of treatment and only address motivational or unconscious elements if they remain unexplained by neurobehavioural origins. Individual treatment plans are designed on a case-by-case basis to capture the individual's unique problems and strengths.

Analysis of authentic and emotionally charged experiences facilitates self-awareness using metaphoric problem identification, followed by strategy design and modification [151]. In group therapy, the therapist acts as an educator and facilitator, assisting with goal identification, the leading theme focused or problem assessment discussion, and offering support and encouragement [76]. Characterisation of ADHD focuses primarily on executive function deficits, followed by inattention and memory. This focus is reflected practically in treatment design as hyperactivity/impulsivity is considered less prevalent in adults [76]. Barkley is a primary citation in all four documents.

*Dialectical Behavioural Therapy (DBT).* Ten studies identified an adapted model of Dialectical Behavioural Therapy (DBT) for ADHD. These consist of randomised controlled trials [21, 153–155], a pragmatic open study [156], and group interventions [157–161]. This treatment model recognises ADHD neurobiological deficits but is grounded in a phenomenological conceptualisation, perceiving the nature of ADHD as a personality disorder. This conceptualisation is supported by similarities in symptoms and the positive response to the treatment of ADHD with comorbid Borderline Personality Disorder (BPD) [158]. Linehan [162] characterises BPD as a disorder of self-regulation from biological irregularities combined with dysfunctional environments, including their interaction and transaction. Experiences of invalidating environments impair childhood ability to learn to label experiences and emotions, modulate emotional arousal, tolerate distress, or form realistic goals and expectations, resulting in a child who invalidates their own experiences, generating a lack of self-trust. The adapted model is presented in group format of 13 weeks of 2-hour sessions. The design prioritises ADHD symptom-oriented modules, highlights non-empirically researched resources of ADHD [163], and includes DBT "mindfulness" training explicitly. The therapist's role in the DBT adapted model for ADHD supports treatment aims of learning to "control ADHD—instead of being controlled by ADHD" through psychoeducation and provision of session structure and flexibility for individuals. A key therapist practice adopted from DBT is the dialectical balance between validating symptoms, aiming for a stabilising effect and encouragement of motivation, and skills training for behavioural change [158]. These models characterise ADHD as a deficit of attention and emotional control with hyperactive and impulsive behaviour, but later papers highlight issues with executive function and self-regulation [157]. Four studies cite Wender [164] as diagnostic criteria [153, 158–160], two studies cite DSM-IV [154, 155], and four studies cite Barkley specifically [21, 156, 157, 161].

*Mindfulness.* Twenty-two documents included mindfulness in treatment options for ADHD. These included systematic reviews [42, 165–167], randomised controlled trials [168–174], a pragmatic open study [156], group interventions [98, 175, 176], a case-control study [177], a multiple case study [178], and psychotherapeutic guidance [8, 179–182]. Only two studies presented mindfulness treatment alone [98, 156]. In Edel et al. [156], mindfulness was used as a comparator to DBT.

Mindfulness-based approaches conceptualise ADHD as a neurobiological disorder of self-regulation with deficits in executive function. Issues with sustained and selective attention are addressed by mindfulness meditation, which is presented as a self-regulatory practice recognised as mental training to strengthen and improve regulation of attention, emotions and brain function [167, 175, 177, 181]. The therapist's role is primarily to introduce and support developing the new skill set of "mindful awareness" or cognitive defusion to facilitate the ability to decrease emotional responses while continuing to act [8]. Interestingly, Zylowska's [175, 176] Mindfulness-Based Cognitive Therapy treatment model includes within its psychoeducation a characterisation of ADHD as a "neurobiological difference" with both evolutionary non-adaptive and potentially adaptive aspects [183–185]. However, within the treatment approach, the ADHD characterisation remains based on cognitive behavioural theory.

Treatment is in a group format, and length varies from 8 to 12 weeks of 2 to 3-hour sessions. The characterisation is reasonably consistent across this group, focusing primarily on poor sustained attention, inhibition and emotional dysregulation attributed to executive dysfunction, with one study highlighting impairments in performance monitoring [173]. Two papers cite DSM IV [42, 168], two cite DSM V [167, 170], and sixteen cite Barkley specifically [8, 98, 156, 165, 166,169, 171–173, 175, 176–180, 182].

**Coaching.** Thirty-six documents presented coaching as a beneficial intervention for ADHD. These include a systematic review [186], a randomised controlled trial [187], individual interventions [188–199], qualitative studies [200–204], quantitative studies [205–207], and psychotherapeutic guidance [118, 143, 208–218]. It is important to note that nineteen studies were conducted at university for students, and therefore have academic goal achievement as a focus [187–195, 197–199, 201, 204, 206, 208, 212].

The term "coaching therapy" was coined by Hallowell and Ratey [143] to highlight the need for a therapist to take a more "active, encouraging role" with ADHD patients. The role of the "therapist-coach" was to provide a structuring force, maintaining focus and reminding patients of goals and objectives through directive interaction, as opposed to open-ended psychoanalysis. ADHD Coaching has since developed into an independent modality, which can be delivered alone or as part of a multi-modal approach. The ADHD Coaches Organisation (ACO) defines ADHD Coaching as a blending of three elements: Life Coaching, Skills Coaching, and Education [218]. Life coaching separates ADHD Coaching from therapy by highlighting the therapist-client relationship's collaborative nature, where the coach supports client self-awareness and achievement of self-identified goals, providing structure and accountability as needed. The client is viewed as a creative and resourceful expert with individual strengths which are leveraged in skills coaching to design systems and strategies to strengthen clients' ability to manage daily life. Education is provided through relevant ADHD research and tools, as requested by the client or as needed.

Conceptualisations of ADHD within coaching models focus almost exclusively on working with neurobiological deficits in executive function, with the primary treatment aim to set and achieve goals and develop skill sets to support practical day to day management. Some models even define themselves specifically as "Executive Function Coaching" [191, 195, 198, 204, 206, 208]. However, some models highlight ADHD Coaching as based on or similar to CBT [186, 196, 199, 208]. The role of the coach is to support clients to improve self-regulation, defined as the ability to persist in goal-directed behaviour through time [204, 209], by modelling cognitive strategies, practising non-judgement, offering pragmatic support and guidance, and holding clients accountable by reflection in session or monitoring progress via between session check-ins. Negative emotions are addressed as barriers to goal achievement and confidence, but models are specific that ADHD coaching is practical [186], dealing with "what, when and how–never why" [213].

Six documents mention self-determination models as part of a wider ADHD Coaching treatment model [190, 191, 194, 195, 204, 206]. These are functional theory models designed to assist students, particularly those with learning disabilities, to develop internal or dispositional characteristics of self-determined behaviour and goal acquisition [219–221]. Field & Hoffman's model [221] defines self-determination as the ability to define and achieve goals grounded in knowing and valuing oneself, which can be supported or thwarted by internal variables and environmental factors. The model specifically focuses on internal controllable variables to assist individuals to adapt to environments with unpredictable support. The core theory is that to be self-determined, one must develop internal awareness and the skills and strength to act on this internal foundation. The model has five major components:

- "Know Yourself": Increase awareness of one's preferences, strengths, weaknesses and needs by "dreaming" or overcoming barriers in socialised expectations for individuals with disabilities that limit options and perceptions of self-efficacy, building on a foundation for self-determined decision making.

- "Value Yourself": Develop affective variables of self-esteem, including identifying strengths in areas commonly perceived as weakness, supporting the self-acceptance of disability and motivation for self-advocacy, increasing the ability to be self-determined.

- Plan: Learn planning skills and visual rehearsal of creative and effective actions for short-range steps leading to long term goals.

- Act: Awareness of how to assertively communicate goals, desires and intentions to others and access relevant resources. Understanding persistence, negotiation, and conflict resolution around risk-taking and barriers that may result from taking action.

- Experience Outcomes and Learn: Learn skills in evaluation of progress based on experience of change and comparison to expected outcomes. Recognition and celebration of successes crystallises the self-determination process.

Wehmeyer et al.'s model [219, 222] is a teaching model to help students become causal agents. Based on cognitive behavioural theory [223, 224], social cognitive theory [225] and research in self-management and self-control [226], this model defines self-determination as the abilities necessary to act as one's primary causal agent and make choices and decisions about the quality of life free from external influence and interference [227]. Developed from a model designed to teach students decision making, independent performance, self-evaluation, and adjustment skills, the updated model includes defining those who are self-determined to persistently regulate problem-solving to meet self-directed personal goals using student-directed learning strategies [219]. This ability is developed through a learned problem-solving sequence of thoughts and actions to reduce the discrepancy between what students want or need and what they have or know. The sequence requires the students to 1) identify the problem; 2) identify potential solutions; 3) identify barriers to solving the problem; and 4) identify consequences to each solution, thereby enabling the student to regulate problem-solving by setting goals to meet needs, constructing plans to meet goals, and adjusting actions to complete plans [219]. A comprehensive combined curriculum of these frameworks was later developed [220]. While they provide support for client autonomy and causal agency within the design of these ADHD Coaching models, these models prioritise goal setting and identification as regulators for human behaviour and recommend student-directed learning strategies based on operant psychology, applied behavioural analysis and positive reinforcement techniques. Thus, treatment approaches for ADHD remain based on cognitive behavioural theory.

Treatment approaches in ADHD Coaching models are primarily cognitive behavioural, including reframing negative self-talk [228], continuous reinforcement [189, 209], implementing rewards and consequences [188, 189, 192, 196, 212], and between-session assignments [192, 196, 209, 217]. These models focus on the characterisation of ADHD as deficits in executive function relating to goal-directed behaviour, disorganisation and planning, motivation, and ultimately self-regulation. Citations for characterisation in ADHD Coaching models include one referencing DSM IV [211], three reference DSM-IV-TR [191, 202, 214], one reference to Brown's Executive Function Model [195], and twenty-eight reference Barkley specifically [118, 128, 143, 186, 188, 189, 191–194, 196–201, 203–207, 209, 212, 214–218].

**Other interventions.** Fourty-seven documents describe non-pharmacological interventions not based on psychotherapy. These include Neurofeedback, Transcranial Stimulation,

Hypnotherapy, Light Therapy, Computer-Based, Mentoring, Self-Monitoring, Binaural Beat Auditory Stimulation, and Movement-related interventions.

*Neurofeedback*. Twelve documents explored Neurofeedback as an intervention for ADHD. These include randomised controlled trials [229–231], individual interventions [232, 233], case-control studies [234, 235], a single case study [236], and treatment guidance [118, 237–239]. Neurofeedback (NF) treatment models focus heavily on neurocognitive deficits as being the origin of ADHD behaviours. The research uses Electroencephalography (EEG) measures to study the correspondences between intracranial electrical currents and responding voltages on the scalp. These measures indicate aspects of brain electrical function and processing, such as the electrical activity of various brain regions and their response to stimuli during cognitive tasks. EEG activity is quantified by computation of amplitude and power values for specific frequency bands of activity, source localization, and brain electrical activity mapping. Frequency refers to the number of oscillations, or waveforms, within a given time period. Analysis of waveforms, or a mixture of frequency bands, is a relational and complex process of examining frequency bands associated with both regions of the brain and cognitive or behavioural characteristics.

Characterisations of ADHD are presented as disturbances in cortical arousal, executive function, and self-regulation. Theta/beta and theta/alpha waveform ratios (TBR) are considered a measure of differences in excess, slow-wave activity and epileptiform spike and wave activity [240], interpreted as abnormal brain processes indicating cortical under arousal, insufficient inhibitory control, and maturational delay in ADHD [241]; however recent studies have challenged TBR as a marker for ADHD diagnosis [235]. Sensory-motor rhythm (SMR) or low beta waveform ratios are thought to indicate cortical hypo-arousal, interpreted as deficiencies in the early stages of information processing [230]. Decreased contingent negative variation (CNV), a steady, slow, negative-going waveform associated with cognitive energy in anticipation of task performance, is considered indicative of dysfunctional regulation of energetical resources in ADHD [234].

Based on research in children, two treatment approaches reflect changes in the conceptualisation of ADHD and, therefore, treatment aims. Traditionally, the focus of treatment has been based on a "conditioning and repair model" [242]. Treatment aims to address dysfunctions and see behavioural improvement and remediation of symptoms following NF application [243]. Skill acquisition and learning are implicit, automatic, and unconscious. Changes in activity indicate positive results: the ability to decrease slow-wave activity (theta) and/or increase fast wave EEG activity (beta) should correlate with symptom improvement; or modulation of slow cortical potentials (SCP), changes of cortical electrical activity, indicate improved cortical regulatory processes [244]. The role of the therapist is to act as a model for affect regulation [236] as well as use behavioural principles such as operant conditioning (i.e., positive reinforcement) in the training process resulting in normalisation and stable change in resting EEG, or "EEG trait" [245], and behaviour [231, 233, 234].

More recently, the NF treatment focus has developed into a "skills acquisition model" [242]. Rather than simply improving neuropsychological deficits, it is thought that NF may be used as a tool for enhancing or optimising specific cognitive or attentional states [246, 247]. This model recognises the bio-psycho-social model of neurodevelopmental disorders, characterising ADHD as impairments in attention, executive functions and self-regulation [229, 230]. In this model, self-regulation, or neuro-regulation, is defined as explicit learning of controlled cognitive processes of cortical regulation evidenced by normalised shifts in EEG amplitudes [242, 248, 249]. Performance optimisation is evidenced by improved skill in changing the "EEG state" via self-initiated effort during task performance [243, 250]. The therapist's role is to use cognitive behavioural therapy elements such as positive feedback and coaching and

operant procedures as active support within treatment sessions to enhance self-efficacy and self-confidence to support neuro-regulation [244, 251]. Citations for characterisation of ADHD in NF models include two citations for DSM-IV-TR [231, 238], two for DSM -V [232, 233], one for Sonuga-Barke's Delay Aversion Model [237], three for Sergeant's Cognitive-Energic Model [230, 234, 237], and four citations for Barkley [118, 229, 237, 239].

*Transcranial stimulation.* Four documents present Transcranial Stimulation as a treatment approach for ADHD. These include a systematic review [252], two randomised controlled trials of Transcranial Direct Stimulation (tDCS) [253, 254], and a randomised controlled trial of Transcranial Magnetic Stimulation (rTMS) [255]. Both forms of transcranial stimulation conceptualise ADHD as a neurobiological disorder with deficits in executive functions, including attention, working memory, impulsivity, and inhibitory control. The treatment aims to increase cortical excitability in the area of stimulation, leading to improved neuropsychological and cognitive functions.

Treatment approaches are non-invasive but differ in their application. Transcranial Magnetic Stimulation uses a coil placed on the subjects head to deliver brief, intense pulses of current (up to 50 Hz) to generate a sizeable electromagnetic induction field initiating neurotransmitter release in the cortex and subcortical white matter of the brain [255, 256]. Transcranial Direct Current Stimulation uses conductive sponge electrodes applied to the scalp in specific locations to deliver a weak electrical current (1–2 mA or milliamps) for up to 20 minutes. It is hypothesised that the electrical current changes the polarisation of the neurons, affecting their average level of discharge [253, 254, 256]. Multiple treatments are administered daily for 3–4 weeks. Protocols suggest two applications of stimulation: "online", or while a patient is completing a task, or "offline" where the treatment is applied before or without specific targeted tasks. Citations for characterisation of ADHD in these models include DSM-IV [252], DSM-IV-TR [254], DSM V [255] and Barkley [253].

*Hypnotherapy.* Two RCTs examined hypnotherapy as a treatment approach for ADHD [74, 257]. These studies conceptualise ADHD as a developmental neurobiological disability with deficits in attention, issues with hyperactivity/impulsivity and problems in executive function, including processing speed, regulating alertness, modulating emotions, and utilizing memory. Treatment aims to improve symptoms, mood, quality of life and cognitive performance. Treatment design is based on symptoms outlined in the DSM-IV and Brown's Executive Dysfunction Model [258]. The therapist's role was to follow a semi-structured manual to review the previous session, present the theme for the current session, perform induction and guided hypnotherapy with a post-hypnotic suggestion, and lead discussion. Treatment length was ten weekly sessions of 40 to 60 minutes. Citations for characterisation of ADHD were the DSM-IV [257] and Brown's Executive Dysfunction Model [74].

*Light therapy.* Five documents present light therapy as a treatment approach for ADHD: a systematic review [259], an individual intervention [260], a quantitative study [261], a literature review [262], and treatment guidance [263]. These documents conceptualise ADHD as a neuropsychiatric disorder with primary symptoms of impulsivity, inattention, and hyperactivity impacted by mood regulation difficulties, maintaining arousal and sleep disturbances that contribute to pathophysiology. This conceptualisation is supported by links between ADHD, seasonal affective disorder (SAD) and circadian rhythms and highlighted by similarities in symptoms between sleep deprivation and ADHD [261, 263]. Research indicates abnormalities in circadian related physiological measures such as heart rate increase relevant to autonomic function, dysregulation in melatonin rhythm leading to delays in melatonin onset, which may affect the modulation of the sleep/wake cycle [263, 264], as well as some evidence of low cortisol impacting wakening times [259]. Also, a later diurnal preference, or evening chronotype, is highly prevalent in the ADHD population. Its association with shorter night sleep periods is

believed to generate sleep debt, delay the sleep phase, and exacerbate symptoms or potentially play a causal role in ADHD symptoms [262, 263].

Light Therapy (LT) treatment aims to assist with phase-shifting abnormal circadian rhythms through light exposure to achieve sleep onset to improve alignment with work, academic, or social norms. Treatment outcomes are improved sleep and improved ability to maintain effort, arousal and attention [260, 262]. The treatment has been trialled as a three-week self-administered daily dose of 10,000 lux at a distance of 24 inches using a full-spectrum fluorescent lightbox [260]. Citations for the characterisation of ADHD in these documents include DSM-IV [260], DSM-V [259], Douglas [262], Brown's Executive Dysfunction Theory [261], and Barkley [261].

*Computer-based interventions.* Eight documents presented computer-based interventions as a treatment approach for ADHD. These include randomised controlled trials [265–268], individual interventions [269, 270], and case-control studies [271, 272]. These approaches characterise ADHD as a neurobiological disorder with executive function deficits, including difficulties in sustained attention, response inhibition, goal persistence, and working memory. Computer-based interventions take two approaches: supportive or training. Supportive interventions aim to target specific symptoms and facilitate functioning via supportive software. Individuals are given access to tools used independently following training for a set timeframe. In Hecker et al. [271], a software tool designed to reduce internal and external distractions aimed to reduce effort and improve engagement, resulting in increased time reading and comprehension. In Irvine [269], a smartphone app for time management aimed to reduce the discrepancy between the perception of time and actual time spent by providing immediate real-time feedback on the current status and time use, leading to adjustments of future tasks according to behavioural therapeutic principles.

Training interventions aim to strengthen cognitive skills and/or remediate deficiencies via cognitive behavioural learning strategies of repetition and positive reinforcement. Working Memory Training [265, 266, 268] aimed to enhance auditory-verbal and visual-space working memory through intensive training with increasing task difficulty leading to improved cognitive and academic performance and attentional self-regulation. Cognitive ability training [272] aimed to improve cognitive skills of decision making, attention, organisation and time management through simulated activities in a gaming environment, providing immediate real-time rewards. Cognitive training for executive function [267, 270] aimed to remediate cognitive processes deficiencies by repeated and graded exposure to neutral and universal stimuli and feedback. Training is self-administered, hierarchical and adjusted to individual performance with outcomes for improvements in daily executive functioning, occupational performance, and quality of life. Treatment length varied in frequency and intensity, from 20-minute sessions 3–5 times a week for 12 weeks to 45-minute sessions five days a week for five weeks and included weekly check-ins or supportive coaching. Citations for characterisation of ADHD in these approaches include DSM-IV [265, 266, 268, 270], DSM V [271], Brown's Executive Dysfunction Model [267, 270], Nigg's Integrative Theory [267], and Barkley [266, 267, 269, 270].

*Mentoring.* One study presented mentoring as an individual intervention for ADHD [273]. Based in a university environment, ADHD is characterised as deficits in basic cognitive skills, such as attention, concentration, and memory and higher-level cognitive skills or "executive functioning", such as planning, organization, judgment, problem-solving, and cognitive flexibility. These can negatively affect the university experience, as more independent self-management and a complex skill set are required for success, particularly time management and organization, academic skills, and social skills.

The mentoring program pairs second-year master's level occupational therapy (MSOT) students (mentors) with undergraduate college students (mentees) for one-to-one support twice weekly for 2-hour sessions in the fall and spring semesters. This mentoring is a credit-bearing course that addresses skill development in time management and organization, academic skills, and social skills for college success. Mentees are graded on attendance, professional behaviours, compliance on a weekly to-do list, a presentation on academic resources, and a 4-part written paper on an academic skill. Mentors are participating as part of a professional Occupational Therapy training programme with an overall goal to facilitate student success in college, and if factors overwhelmingly interfere with this goal, to identify an alternate, suitable plan. As part of the training, mentors meet in discussion groups to brainstorm ways to overcome the mentoring process's challenges. The citation for the characterisation of ADHD in this intervention is primarily the DSM V [273].

*Self-monitoring.* One study presented individual self-monitoring as an intervention for ADHD [274]. Based in a university environment, ADHD is characterised as a neurobehavioral disorder with symptoms of inattention, hyperactivity, and impulsivity, which increases the risk of academic failure or underachievement.

A checklist tool is co-designed and supported with integrity checks and email reminders every 2–4 days, with face-to-face check-in sessions every two weeks. The self-monitoring intervention aims to teach participants to observe and record behaviours to change the behaviour in the future. Outcomes are to obtain higher grades, endorse fewer ADHD symptoms, engage in more positive study skills, further attain goals, and improve medication adherence. Citations for characterisation of ADHD is DSM IV [274].

*Binaural beat auditory stimulation.* Two documents present binaural beat auditory stimulation as a treatment for ADHD. These include an individual intervention [275] and a literature review [276]. These approaches characterise ADHD as a disorder with core deficits in behavioural inhibition and sustained attention, highlighting a decrease in beta wave states interfering with maintenance of attention as a contributing factor.

Binaural beat auditory stimulation generates tones of two frequencies presented separately in each ear which are synthesised by the medulla into a single low-frequency tone. The pulse frequency from this binaural beat is the difference between the two tones and generates electrical activity that EEG can record. Treatment aims to match the difference between the tones to a particular brain-wave state, such as the beta range, which will correspondingly be maintained by overall brain activity and affect cognition levels [277]. Treatment involves exposure to auditory stimulus via headphones during an active task. Citations for characterisation of ADHD only directly reference Barkley [275].

*Movement-related interventions.* Twelve documents present movement-related interventions as a treatment for ADHD, including a systematic review [278], a pilot study [279], case-control studies [280–285], and treatment guidance [286–289]. In these approaches, ADHD is a disorder with core issues in special working memory, attention control, response inhibition, motor control, delay aversion, emotional self-regulation, and executive dysfunction. Movement-related interventions approach treatment in two ways: passive and active.

One document presented a passive intervention. Whole Body Vibration (WBV) devices deliver sinusoidal or oscillating wave vibrations at low frequencies to enhance mechanical muscular performance [290], improve balance and proprioception [291], and increase vigilance [292], potentially by inducing muscle contractions and increasing tension through the stretch reflex. Treatment is passive, delivered while sitting still, and aims to improve attention, inhibitory control, and cognitive performance in ADHD [280].

Active movement-related interventions aim to improve neurobiological factors such as increased cerebral blood flow, enhance neuroplasticity [288, 289], assist the development of

cortical and subcortical brain regions through activity [287], reduce the impact of comorbid anxiety, depression, stress and negative affect [279, 288], and improve cognitive function and performance [282–286]. There is a specific focus on hypodopominergic functioning in ADHD and the upregulation of a brain-derived neurotrophic factor (BDNF) protein in several studies. [281, 283, 286, 288, 289]. Research shows that BDNF is linked to differentiation and survival of dopaminergic neurons, and decreased levels of BDNF have been suggested as being involved in ADHD pathology [293]. As well as improved cognition, one of the benefits of acute exercise is elevated levels of BDNF, which these models argue makes exercise an important intervention for ADHD. Treatment varies both in approach and length, from vigorous physical activity for 30 minutes, such as cycling, to fine motor movement stimulation using an anti-stress ball during a task. Citations for the characterisation of ADHD in these approaches include DSM IV [287], DSM V [285], Nigg [281, 286], Sergeant [282], Sonuga-Barke [282, 286], and Barkley [278–280, 283, 284, 286, 288, 289].

**Alternative models.** *Psychoanalysis and Psychodynamic.* There are very few studies in Psychoanalysis and Psychodynamic approaches for adult ADHD. A group intervention [294], single [295, 296] or double case studies [297–299] were reviewed, as well as an evaluation study [300]. Much of the literature consists of literature reviews [35, 301–304] and guidance pieces [152, 305, 306], which demonstrate considerable debate in the characterisation and aetiology of ADHD. Early papers reflect issues in clinical approaches by highlighting the importance of considering ADHD diagnosis as defined by DSM-IV in light of epidemiological evidence [301, 307]. Both Psychoanalysis and Psychodynamic approaches present alternative models to Barkley, with distinct variation in characterisation.

Historically, Psychoanalysis does not recognise neurobiological deficits. Behaviours associated with ADHD are conceptualized as disturbances in the ego, identified as the organising force responsible for synthesis and integration of internal and external stimuli, internalisation of object relations and structure and development of the superego, and integral to facilitating the capacity for self-observation and self-reflection. Early presentations of these disturbances in childhood lead to attachment issues and interfere with sibling relationship development [301]. Behaviours are perceived as defence mechanisms, identified as an internal struggle for control [296, 300]. Psychodynamic perspectives differ in that behaviours are conceptualized as a reaction to neurobiological deficits [152, 302, 306], facilitating engagement with Barkley's model. Executive functioning deficits are presented as synonymous with self-regulation deficits, interfering with the development of personality structure and an internal representation of self about others. Self-regulation deficits disrupt the ability to empathise, which distorts the capacity to mentalise and develop a coherent sense of self [304].

The therapist's role in these models is to act as the organising force for the client, assisting them to develop ego capacities via therapeutic relationship and transference. This enables the client to experience empathy, recognise mental states, and identify self in relation to others [35, 301, 304]. With the exception of the group intervention [294], treatment designs are intensive, up to four times a week [301, 304] and long term, between 2 and 12 years [295, 296, 298, 301, 304]. Despite the alternative model to characterise ADHD, four studies reference international guidance [297, 301, 302, 307], seven studies mention executive function or cognitive control [35, 152, 295, 297, 299, 304, 306], and seven reference Barkley specifically [35, 294, 295, 297–299, 304].

## Conclusion

A review of 221 documents confirmed that treatment approaches for ADHD are based on a dominant cognitive behavioural paradigm for conceptualising ADHD, which attributes

symptoms solely to neurobiological and developmental deficits leading to challenges with cognitive function, behavioural control, and management of self-regulation. This is reflected in descriptions of treatment aims, approaches and outcomes (S1 Appendix).

While this scoping review aimed for as broad a scope as possible, it is important to acknowledge the limitations of this study. First, while translation services were used as much as possible, the material identified in the results were primarily published in English. Further, the majority of the documents presented were published in the US, Canada and European countries. This may be due to documents being presented or published by journals not listed by the major search engines, and therefore not identified in the search strategies. Alternatively, there may not be a large existent body of published research in other countries, as the official diagnosis criteria for adults with ADHD was only recognised in 2013 [149]. Secondly, this scoping review was an enormous undertaking, and results are only up to date as of April 2020. However, searches did not reveal any other recent reviews of the theoretical charactarisation of ADHD, therefore it is believed this is the most current comprehensive scoping review on the topic.

This review reflects current research understanding that ADHD is complex and multidimensional in its presentation and impact. Clearly, it shows a broad, cross-disciplinary interest in developing treatment approaches to support individuals with ADHD to reduce symptoms, improve functioning and achieve a better quality of life. Critically, it highlights that a single theoretical perspective limits research into effective treatments for ADHD. Existing aetiological theories of ADHD have been challenged for their refutability [308], and other issues such as accounting for context variability, or inability to fully link or account for the full aspects of the symptomology [19–21], and heterogeneity [1, 22–24] including specific links between domains and outcome [22] and cognition and motivation to select actions for a given context [309]. Recent recommendations for resolving challenges with heterogeneity in ADHD emphasise the importance of theoretical guidance in decision-making and recognise the critical role of beliefs, assumptions, and goals in preventing misapplication of conclusions to clinical circumstances or populations [1]. It is proposed that treatments based on approaches from a singular perspective on processes of self-regulation and a deficit-based origin of impairments in ADHD may be limited in scope and capacity to identify and support positive psychological factors for well-being and growth. Hence, the findings in this scoping review identify a gap in research and practice for alternative theoretical perspectives of ADHD.

This review concludes that further research into additional theoretical models of self-regulation would provide opportunities to develop alternative treatment approaches and benefit research and understanding of the symptomology of ADHD.

## Supporting information

**S1 Appendix. Analysis of treatment approaches.**
(DOCX)

**S1 File. PRISMA scoping review checklist.**
(PDF)

## Author Contributions

**Conceptualization:** Rebecca E. Champ.

**Data curation:** Rebecca E. Champ.

**Formal analysis:** Rebecca E. Champ.

**Supervision:** Marios Adamou, Barry Tolchard.

**Writing – original draft:** Rebecca E. Champ.

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
