## [Decision Letter · Decision Letter 0]

18 Oct 2021

PONE-D-21-16848

The Impact of Psychological Theory on the Treatment of Attention Deficit Hyperactivity Disorder (ADHD) in Adults: A Scoping Review

PLOS ONE

Dear Dr. Champ ,

Thank you for submitting your manuscript to PLOS ONE. After careful consideration, we feel that it has merit but does not fully meet PLOS ONE’s publication criteria as it currently stands. Therefore, we invite you to submit a revised version of the manuscript that addresses the points raised during the review process.

Kindly address all points raise dby the two reviewers and resubmit by November 15th, 2021. If you need more time, you can inform the journal office at plosone@plos.org. Please include the following items when submitting your revised manuscript:

We look forward to receiving your revised manuscript.

Kind regards,

Gerard Hutchinson, MD

Academic Editor

PLOS ONE

2. We note that this manuscript is a scoping review; our author guidelines therefore require that you use PRISMA guidance to help improve reporting quality and consistency for this type of study. Please include copies of the completed PRISMA checklist for scoping reviews (http://www.prisma-statement.org/Extensions/ScopingReviews) as Supporting Information with a file name “PRISMA checklist

Additional Editor Comments (if provided):

Reviewers' comments:

Reviewer's Responses to Questions

**Comments to the Author**

1. Is the manuscript technically sound, and do the data support the conclusions?

Reviewer #1: Partly

Reviewer #2: Yes

2. Has the statistical analysis been performed appropriately and rigorously? 

Reviewer #1: N/A

Reviewer #2: N/A

3. Have the authors made all data underlying the findings in their manuscript fully available?

Reviewer #1: Yes

Reviewer #2: Yes

4. Is the manuscript presented in an intelligible fashion and written in standard English?

Reviewer #1: Yes

Reviewer #2: Yes

5. Review Comments to the Author

Reviewer #1: Authors make a very good review on characterizations of ADHD from different psychological approaches, as well as the different psychological therapies applied to patients. Description of psychological approaches and characteristics of therapies applied is detailed and sufficient. In this regard, it is an interesting article on the history of ADHD psychological treatment. However, the present manuscript is largely descriptive, and could be greatly improved by adding information that would also be useful in the discussion section, as well as to support authors´conclusions.

MAJOR ISSUES

1.- The hierarchy of ideas is not always clear, it would benefit from establishing headings and subheadings clearly. For example, at first authors talk about behavioral therapy and state that the review will address this subject in waves, and describe the first wave in page 7, line 150 (authors could use a subheading for this section). However, after first wave authors address REBT (page 8, line 167), and later return to the second wave on page 8, line 181). Are authors considering REBT a subheading of the CBT first wave? This situation repeats along the manuscript. Please add subheadings as needed.

2.- In the present version of the manuscript, a detailed description of each psychological approach is provided, along with information regarding the specific therapeutic interventions. However, authors do not state whether the interventions they mention (first wave, REBT, second wave, etc) were successful or not (except for light therapy and computer based interventions, where some treatment outcomes are mentioned), or which symptoms were treated more effectively, or if follow up studies revealed information regarding symptoms control in the long term, or if sessions could be separated for longer periods of time as treatment advanced, etc. Information regarding the effectiveness of each of the psychological therapies addressed would be necessary to understand not only how psychological approaches have changed, but also if these changes had an effect on patients´ health. Please include information regarding effectiveness of the psychological treatments described, it could be included as a paragraph at the end of each section (first wave, second wave, etc.).

Furthermore, it would be useful for discussion. In page 24, lines 633 and 634, authors say that “Critically, it highlights that a single theoretical perspective limits research into effective treatments for ADHD”, which implies first, that current psychological treatments for ADHD can be more effective, and secondly, that there are psychological approaches, currently not considered, that could improve treatment effectiveness. Therefore, this statement should be supported by at least two points in the discussion section: a) Specify how the effectiveness of psychological treatment could be improved, i.e. which outcomes could be better (improvement in more areas, observing effects after a shorter treatment, etc); and b) propose some aspects not considered in the current psychological approach, that could be useful when addressing ADHD, improving treatment effectiveness.

MINOR ISSUES

1.- Page 3, Line 65. Quebec is not a language

2.- It would be very useful if authors included a table comparing the different theoretical approaches for ADHD, outlining similitudes and differences

Reviewer #2: The main aim of this review was to consider the use of psychological theory in the development of ADHD treatment for adults. Key themes of the cognitive-behavioural approach to adult ADHD have been explored. The review is a useful contribution to the field. Whilst literature is available on the topic, I would suggest this is a timely update that explores a breadth of evidence.

The rationale for the scoping review is clear. It is written clearly and easy to follow. The review is well informed and provides the field with a useful foundation for future research directions.

I would suggest minor revisions would be beneficial to the reader, the sentence ‘characterisations of ADHD that were not empirically researched’ on line 98, page 5 could be expanded on, to let the reader understand the inclusion criteria further. Also, attention should be paid to the reference list, some information is missing/inconsistent due to formatting errors.

6. PLOS authors have the option to publish the peer review history of their article (what does this mean?). If published, this will include your full peer review and any attached files.

Reviewer #1: No

Reviewer #2: No

---

## [Author Response · Author response to Decision Letter 0]

16 Nov 2021

I would like to thank the reviewers for their time and comments. The following details the response to requests for revisions:

Reviewer 1:

The reviewer raises excellent points about the manuscript. As requested, the hierarchy of ideas has been clarified and headings and subheadings are more specific. 

However, their comments regarding efficacy and further detail in the discussion address an entirely different focus and aim than the current piece presents. I agree the request for detailed information on the efficacy of treatments throughout the manuscript would assist the reader to “understand not only how psychological approaches have changed, but also if these changes had an effect on patients´ health”, if this were an article focused on the history and development of the efficacy of treatments for ADHD as Reviewer 1 describes it. However, the focus of the manuscript is to identify, map and confirm the theoretical underpinnings of the characterisation of ADHD which informs treatment design, approaches and outcomes.

Perhaps this is less clear because the questions that prompted the research had not been included. Therefore, additional information is provided in the introduction which will clarify the aims of the research (lines 59-64). I have also addressed the question of evidence for efficacy in treatment of specific symptoms (lines 40-45). 

I am very pleased to receive comments regarding requests for a comparative table of psychological theories, and information on “how the effectiveness of psychological treatments could be improved” and to “propose some aspects not considered in the current psychological approach”. This is a much larger and more in-depth question that goes beyond the current manuscript. A second manuscript directly addressing these queries has been submitted for publication and is currently under review. Therefore, I have now described issues with current treatment approaches as highlighted in recent systematic reviews (lines 45-51) but overall the current work only addresses these queries nominally (lines 52-58 and 673-678) to prepare the ground for a future discussion of the kind that Reviewer 1 has requested.

Reviewer 2:

I appreciate that some readers may not be aware of non-empirical characterisations of ADHD. Therefore, I have defined this body of literature more clearly, provided examples, and given further explanation for their exclusion (lines 113-119). I have also reviewed and updated my references.

---

## [Decision Letter · Decision Letter 1]

26 Nov 2021

The Impact of Psychological Theory on the Treatment of Attention Deficit Hyperactivity Disorder (ADHD) in Adults: A Scoping Review

PONE-D-21-16848R1

Dear Dr. Champ 

We’re pleased to inform you that your manuscript has been judged scientifically suitable for publication and will be formally accepted for publication once it meets all outstanding technical requirements.

Kind regards,

Gerard Hutchinson, MD

Academic Editor

PLOS ONE

Additional Editor Comments (optional):

Reviewers' comments:

Reviewer's Responses to Questions

**Comments to the Author**

1. If the authors have adequately addressed your comments raised in a previous round of review and you feel that this manuscript is now acceptable for publication, you may indicate that here to bypass the “Comments to the Author” section, enter your conflict of interest statement in the “Confidential to Editor” section, and submit your "Accept" recommendation.

Reviewer #1: All comments have been addressed

2. Is the manuscript technically sound, and do the data support the conclusions?

Reviewer #1: Yes

3. Has the statistical analysis been performed appropriately and rigorously? 

Reviewer #1: N/A

4. Have the authors made all data underlying the findings in their manuscript fully available?

Reviewer #1: Yes

5. Is the manuscript presented in an intelligible fashion and written in standard English?

Reviewer #1: Yes

6. Review Comments to the Author

Reviewer #1: The additions on the revised version of the manuscript allow a clearer definition of its purpose, which is more focused on the theoretical aspect of ADHD characterization and, as stated in the title, its impact on ADHD treatment. In this regard, the first part of these objectives (theoretical aspect of ADHD characterization) is fully addressed along the manuscript. Moreover, what the title implies regarding the impact of psychological theory on ADHD treatment is briefly addressed at the discussion section.

7. PLOS authors have the option to publish the peer review history of their article (what does this mean?). If published, this will include your full peer review and any attached files.

Reviewer #1: No

---

## [Editor Report · Acceptance letter]

2 Dec 2021

PONE-D-21-16848R1 

The Impact of Psychological Theory on the Treatment of Attention Deficit Hyperactivity Disorder (ADHD) in Adults: A Scoping Review 

Dear Dr. Champ:

I'm pleased to inform you that your manuscript has been deemed suitable for publication in PLOS ONE. Congratulations! Your manuscript is now with our production department. 

Kind regards, 

on behalf of

Dr. Gerard Hutchinson 

Academic Editor

PLOS ONE